

# Drought reduces tree growing season length but increases nitrogen resorption efficiency in a Mediterranean ecosystem

Raquel Lobo-do-Vale[1], Cathy K. Besson[2], Maria C. Caldeira[1], Maria M. Chaves[3], João S. Pereira[1]

[1]Centro de Estudos Florestais, Instituto Superior de Agronomia, Universidade de Lisboa, Tapada da Ajuda, 1349-017 Lisboa, Portugal
[2]Instituto Dom Luiz, Centro de Geofísica da Universidade de Lisboa, Faculdade de Ciências, Campo Grande, 1749-016 Lisboa, Portugal
[3]Laboratório de Ecologia Molecular, ITQBNOVA, Universidade Nova de Lisboa, Av. da República, 2780-157 Oeiras, Portugal

*Correspondence to*: Raquel Lobo-do-Vale (raquelvale@isa.ulisboa.pt)

**Abstract.** Mediterranean ecosystems are hotspots for climate change, as the highest impacts are forecasted for the Mediterranean region, mainly by more frequent and intense severe droughts. Plant phenology is a good indicator of species' responses to climate change. In this study, we compared phenology of cork oak trees (*Quercus suber*), an evergreen species, over two contrasting years, including the most severe drought (2005) since records exist. We evaluated not only the timing of occurrence of the vegetative phenophases in spring (bud development, budburst, shoot elongation, trunk growth and leaf senescence), but also their duration and intensity. We also quantified nitrogen in green and senescent leaves and assessed the nitrogen resorption efficiency. Temperature was the main driver for budburst. Nevertheless, water had a main role constraining all the other phenophases, by strongly reducing the growing season length (-48 %) and consequently tree growth. Basal area increment was the most affected growth variable (-38 %), although the rate of increase remained similar among years. Shoot elongation was reduced by -21 % yet elongation occurred at a higher rate in the dry as compared to the mild year. Leaf senescence during the bulk period was higher in the dry year, in which leaves were shed at the same rate over a longer period. Nitrogen concentration in green and senescent leaves were affected by drought and cork oak remarkably increased the nitrogen resorption efficiency (+22%), which appears to be an adaptive trait that mitigates the limitation in nitrogen uptake by the roots during drought. Water availability was the main driver of the growing season length in this Mediterranean ecosystem, although it may also be affected by complex interplays between precipitation and temperature. Our results highlight the importance of studying different phenological metrics to improve our understanding of the ecosystems responses to climate change. The faster dynamics observed in shoot elongation, in contrast with the other phenophases, are indicative that cork oak privileges leaf area development, while all other phenophases develop at the same rate. Finally, a higher nitrogen resorption efficiency in response to drought may clearly improve tree fitness in the short-term, but will probably exert a negative feedback on the nitrogen cycle in the long-term which might affect the ecosystem functioning under the forecasted droughts.





## 1 Introduction

Plant phenology is an important indicator of the ecosystems responses to climate change (Jeong et al., 2011; Polgar and Primack, 2011; Richardson et al., 2013; Han et al., 2018). Shifts in plant phenology have broad consequences for ecosystems, not only directly through changes in the cycling of carbon, water and nutrients, but also indirectly through feedbacks on the
climate system (Diez et al., 2012; Richardson et al., 2013; Tang et al., 2016; Han et al., 2018), which in turn may strongly influence ecosystem productivity, as well as other ecosystem services (Tang et al., 2016). However, the response of species phenology to climate change is complex and highly variable (Penuelas et al., 2004; Polgar and Primack, 2011; Diez et al., 2012; Tang et al., 2016; Gerst et al., 2017), which points to the need for a better understanding of the patterns and climate drivers of the different phenophases in a broader range of tree species (Polgar and Primack, 2011; Richardson et al., 2013;
Doblas-Miranda et al., 2015; Tang et al., 2016; Lempereur et al., 2017). In particular, there is a dearth of information on Mediterranean schlerophyllous species (Misson et al., 2011; Bussotti et al., 2014; Costa et al., 2016) that grow in regions where the impacts of climate change will be amplified, mainly by more severe and longer droughts (Giorgi and Lionello, 2008; Jacob et al., 2014; Perkins-Kirkpatrick and Gibson, 2017).

Temperature is clearly one of the most important drivers of plant phenology (e.g. Pinto et al., 2011; Flynn and Wolkovich,
2018). An earlier budburst and a lengthening of the growing season in Mediterranean and other ecosystems over the globe, due to increased temperatures, have been reported by many studies (e.g. Menzel and Fabian, 1999; Peñuelas et al., 2002; Gordo and Sanz, 2005; Menzel et al., 2006; Richardson et al., 2006; Gordo and Sanz, 2009; García-Mozo et al., 2010; Jeong et al., 2011; Bigler and Bugmann, 2018). However, it was hypothesized that changes in precipitation may be more important than changes in temperature in affecting the phenology of Mediterranean plant species (Penuelas et al., 2004). Still, available
information is scarce. Few studies indicated that precipitation can have an impact on the onset of some phenophases, such as flowering and fruiting (Penuelas et al., 2004; García-Mozo et al., 2010; Swidrak et al., 2013). Nevertheless, for tree growth and overall ecosystem productivity, summer drought is the most limiting period (Niinemets, 2010; Lempereur et al., 2017) and the overwhelming importance of precipitation over the growing season is clear (e.g. Pereira et al., 2007; Oliveira et al., 2016). Mediterranean trees adjust their vegetative activity to take advantage of the most favourable periods of the year, temperature
and water related, which is mainly in spring (Gordo and Sanz, 2010; Misson et al., 2011; Pinto et al., 2011). Studies have shown both total and seasonal precipitation to be important for tree growth. For example, Besson et al. (2014) and Oliveira et al. (2016) emphasized the importance of spring precipitation. Costa et al. (2002) suggested that winter precipitation enhanced early spring growth while Costa-e-Silva et al. (2015) reported reductions in stem growth due to a dry winter. Finally, Costa et al. (2016), comparing cork growth on different sites with different local environmental conditions, identified different
sensitivities to climate and concluded that cork oak strategies to cope with drought range from drought tolerance to drought avoidance. The effect of temperature on tree growth seems to be tightly linked to the precipitation (Sardans and Peñuelas, 2013; Oliveira et al., 2016), stimulating growth if warmer temperatures occur in wetter years and having no effect or decreasing growth in dry years. However, it is not clear how these two climatic drivers affect the timing and length of other tree



phenophases. For most deciduous and evergreen species, the growing season length depends on the timing of budburst in the spring but also on leaf senescence in autumn. For example, Jeong et al. (2011), reporting changes in phenology on the northern hemisphere suggested that the lengthening of the growing season was due to extended leaf senescence in autumn rather than an earlier budburst. But, Han et al. (2018), reported both an earlier spring growth and an extended leaf senescence, in European forests. In the Mediterranean region, Gordo and Sanz (2009) reported that an extended growing season was due to an earlier budburst in the spring, in a study with 29 perennial species.

Mediterranean soils are usually poor in nutrients (Sardans and Peñuelas, 2013). Shifts in plant phenology can affect the cycling of nutrients that in turn might affect the nutrient availability for plant growth (Richardson et al., 2013). In addition, drought can condition nutrient availability for plant growth, either by decreased litterfall decomposition due to limited microbial activity and/or by limited nutrient uptake by roots due to lower mass flow and diffusion. Retranslocation of nutrients from senescent to new leaves is a key process that contributes for the nutrient conservation in plants, reducing the dependence of plants on soil nutrient availability (Aerts, 1996; Fioretto et al., 2003; Xu et al., 2017; Wang et al., 2018). Nutrient retranslocation typically increases in the Mediterranean tree species before leaf senescence (Sardans and Peñuelas, 2013). Climate is a primary driver of nitrogen resorption efficiency (i.e. percentage of nitrogen that is retranslocated or resorbed, from senescent to green leaves, NRE) (Aerts, 1996)., which is known to globally decrease with mean annual precipitation and mean annual temperature (Vergutz et al., 2012; Wang et al., 2018). However, it is not clear if NRE depends directly on annual precipitation or if the reduced water availability elicits a response in NRE (Brant and Chen, 2015).The reported impacts of drought are not consistent. Studies with herbaceous species reported drought to either increase (Zhao et al., 2017) or decrease NRE (Drenovsky et al., 2012; Khasanova et al., 2013). Also, no changes in NRE were observed between two distinct rainfall zones in Australian schlerophyll species (Wright and Westoby, 2003). More importantly, in Mediterranean ecosystems, it is not known how NRE is affected by increasing drought conditions (Sardans and Peñuelas, 2013).

Due to a plethora of adaptation strategies to withstand long periods of water shortage (see a review by Sardans and Peñuelas (2013)), among other particularities, Mediterranean ecosystems are considered exceptional models to study the effects of climate change (Niinemets and Keenan, 2014; Doblas-Miranda et al., 2015). One important Mediterranean tree species is cork oak (*Quercus suber*). It is an evergreen tree with a leafing phenology, characterized by short-lived leaves that usually are shed within one year (Natividade, 1950; Pereira et al., 1987; Escudero et al., 1992; Aronson et al., 2009), concurrently with spring growth. Like other Mediterranean oaks, cork oak is adapted to the Mediterranean climate, being able to survive summer drought. However, this tree species is showing signs of increasing vulnerability, mostly due to increased drought occurrence and poor management practices (Costa et al., 2011; Camilo-Alves et al., 2013). Cork oak phenological responses to the forecasted increase in severe drought episodes are still largely unknown. In addition, available studies on cork oak rarely compared nutrient dynamics in green and senescent leaves (Delarco et al., 1991; Andivia et al., 2009), making difficult to evaluate the impact of climate change, or drought in particular, on nutrient cycling in Mediterranean cork oak woodlands.





In this study, we discuss the phenophases that constitute the cork oak vegetative spring growing season (bud development, budburst, shoot elongation, leaf senescence and trunk growth) in two contrasting years, one being a naturally occurring extreme drought. Each phenophase is characterized by timing, duration and growth. Also, the nitrogen resorption efficiency is determined and compared between the mild and extreme dry year. In this way, we aimed to assess the effect of the extreme

dry year on the cork oak spring phenology and growth. More specifically, we hypothesised that drought would 1) affect the timing of the different phenophases; 2) decrease the length of the spring growing season negatively affecting growth and; 3) increase the nitrogen resorption efficiency from senescent to new leaves.

## 2 Materials and methods

### 2.1 Study site

The study was conducted at Herdade da Mitra (N 38º31.664', W 8º01.380', 221 m altitude) in southern Portugal. The climate is typically Mediterranean, with hot and dry summers. The soil is sandy (88.9 % sand, 4.9 % silt, and 6.3 % clay), with a low water holding capacity (~5 %) and slightly acid (pH 4-6), on a 5 % slope. The experimental area was 0.3 ha. Cork oak (*Q. suber*) trees were 16 years old. Tree density was 1997 ± trees ha$^{-1}$. The understory was mostly composed by *Cistus salviifolius*

and *C. crispus* shrubs, and C3 annual herbaceous species, ending their cycle typically at the beginning of the summer period (June). Spring phenology and growth was monitored over two years (2004 and 2005) in 15 trees under natural conditions. Mean tree height at the beginning of the study was 5.0 ± 0.2 m.

We used the experimental set-up of a water manipulation study, as described in Besson et al. (2014), in which 3 treatments were applied. Dry, Ambient and Wet treatments received, respectively 80, 100 and 120% of annual precipitation. Since rainfall

exclusion had hardly any impact on cork oak physiology, we pooled the data from Dry and Ambient treatments. For this study we used only trees that were sampled for both phenological and physiological measurements (n = 15). This means that we discarded three trees from the above-mentioned sample. For every variable presented in this study, a previous check for treatment effects was performed, i.e., that no treatment effect was observed, to confirm that the pooling was correctly applied.

### 2.2 Environmental monitoring

Precipitation (ARG100 rain gauge, EM Ltd., Sunderland, UK), air temperature (*T*) at 2 m height (Thermistor M841, Siemens, Munich, Germany), photosynthetically active radiation (PAR, LI-190 quantum sensor, Licor Inc., Lincoln, NB, USA) and air humidity (Fischer 431402 sensor, K. Fischer GmbH, Drebach, Germany) were measured in 5 min intervals and 30-min averages recorded with a datalogger (DL2e, Delta-T Devices Ltd., Cambridge, UK). Daily mean temperatures were used to

calculate the thermal time needed for budburst, or degrees-day sum (DDS), *i. e.*, the accumulated temperature above a certain





base temperature during a certain period of time (Cannell and Smith, 1983). We considered DDS from the 1st of January, assuming bud dormancy break (Sampaio et al., 2016), until budburst date, for each year. Base temperature was set to 6.2 ºC, in agreement with the threshold estimated for cork oak in a previous study in the same region (Pinto et al., 2011). To better characterize interannual variations in climatic conditions, our field meteorological data were compared to a reference period

(1971-2000, (IPMA, 2018)).

Results are presented as hydrological years, as they better represent the seasonality of water availability for tree functioning and growth. The hydrological year of 2004 is comprised between October of 2003 and September 2004 and the hydrological year of 2005 between October 2004 to September 2005.

**2.3 Tree phenology and growth measurements**

Phenological observations were performed in one randomly marked apical branch, sun-exposed (south-facing), from a total of 15 trees, at least once a week. We registered the days of bud activation (when bud was visible) and budburst (when shoot emerged). We also registered shoot elongation (when first leaves unfolded and could be count) and trunk growth onset and cessation, and leaf senescence. Then, we determined bud development duration (as the number of days since bud activation to

budburst), shoot elongation and trunk growth duration (as the number of days since the onset to cessation of shoot elongation and trunk growth). The spring growing season length was defined as the number of days since bud activation until trunk growth cessation.

Shoot elongation was measured since budburst with a ruler and leaves counted. Trunk diameter at breast height (dbh) was measured from October 2003 with tree dendrometers (I-802-D1, UMS GmbH, Munich, Germany), at the same time of the day

and at least once a month during spring, and basal area increment (BAI) determined. Since trunk growth in cork oak can be an almost continuous process, depending on climatic conditions (Costa et al., 2002), the onset and end of the spring growth period was defined when dbh increments were lower than 0.05 ‰ day$^{-1}$.(relative to the total tree dbh). BAI was expressed as percentage of increment relative to basal area at the onset of spring growth, of each tree in each year. Autumn BAI was used to confirm the end of the growing season of the dry year, because the measurement performed at the end of July was

inadvertently lost (Fig. S2a). Spring and autumn BAI were regressed against the day of year (DOY).

Leaf area index (LAI) was measured at the end of the summer, as described in Besson et al. (2014). Specific leaf area (SLA), calculated as the ratio of leaf area to dry weight, was determined from 6 discs (with 0.7 cm diameter), collected from three leaves, at least, in each tree, at the end of the growing season.

For litterfall determination, six sets of three litter traps (0.15 m$^2$ each) were randomly placed in areas including at least two of

the studied trees (n = 6). Litterfall was collected at least once per month, depending on the intensity of fall observed, from 13 January 2004 until the end of the study period. Litterfall was separated in leaves and branches, oven dried at 80 ºC for 48 h



and the dry weight of litter leaves obtained. As litterfall was collected in short time intervals, litter leaves were actually senescent leaves. Here, only this litterfall fraction was reported and named as senescent leaves (in g leaves per m$^{-2}$ ground). The bulk period of leaf senescence was considered when fall rate was consistently higher than 0.5 % day$^{-1}$. Senescent leaves were then used for N content determination at the onset and at the end of the bulk leaf senescence period, indicating,

respectively, minimum and maximum N content in senescent leaves for each year. N determination is described below.

To evaluate the intensity of occurrence of the different phenophases, the rate of shoot elongation (cm day$^{-1}$), trunk growth (cm$^2$ day$^{-1}$) and leaf senescence ((g m$^{-2}$) day$^{-1}$) were calculated at the end of the growing period.

## 2.4 Physiological measurements

To monitor tree water status, leaf water potential was measured at predawn ($\Psi_{pd}$) in three south-exposed mature leaves in each of the 15 trees. Measurements were performed periodically over the studied period, with a Scholander-type pressure chamber (PMS Instrument Co., Corvallis, Oregon, USA). To monitor carbon assimilates potentially allocated to growth, daily courses of photosynthesis were measured at 10:00, 13:00 and 16:00 in south-exposed leaves of the selected 15 trees, in same days than $\Psi_{pd}$, in order to assess the maximal rate of photosynthesis ($A_{max}$). Measurements were performed with two cross-calibrated

portable photosynthesis systems (Li-6400; Licor Inc., Lincoln, Nebrasca, USA).

## 2.5 Nitrogen resorption efficiency

Green leaves were sampled periodically, 10 from each of the 15 trees and frozen in liquid nitrogen before being freeze-dried. Green and senescent leaves were ground in a laboratory mill for later analysis of nitrogen (N) content. Nitrogen content was

determined in green ($[N_{gr}]$) and senescent ($[N_{se}]$) leaves by near infrared reflectance spectral analysis (NIR), according to Joffre et al. (1992). Since six samples of $[N_{se}]$ were obtained, $[N_{gr}]$ data from the 15 trees was polled according to the distribution of litter traps (n=6). Nitrogen resorption efficiency (NRE) was determined according to Finzi et al. (2001), Eq. (1), at the onset and at the end of the bulk litter fall period:

NRE (%) = ($[N_{gr}]$ – $[N_{sl}]$)/$[N_{gr}]$ x 100,                (1)

where $[N_{gr}]$ was N content in green leaves and $[N_{se}]$ was N content of senescent leaves.



## 2.5 Statistical analysis

We used paired t-test or the non-parametric Mann-Whitney test, when test assumptions were not verified, to compare the timing and duration of the different phenophases and growth variables between years. To evaluate the effects of drought on N content in green and senescent leaves and on NRE, a two-way repeated measures ANOVA was performed, as the same trees

were repeatedly sampled over the study period, with years (mild and dry year) and phenological stage (onset and cessation of leaf senescence, thus maximum and minimum N concentration in senescent leaves) as factors. The post-hoc Student-Newman-Keuls test was performed when significant factor effects were found. To evaluate the physiological status over the growing season, a F-test for comparison between slope and intercepts was performed between the mild and dry year. The Spearman correlation coefficient was used to evaluate the relationships between variables and, when adequate, linear regressions were

obtained. All statistical analysis were carried out with IBM SPSS Statistics 24 and considered significant when p-value <0.05. Results are presented as mean ± SEM.

## 3 Results

### 3.1 Environmental conditions

Total precipitation and distribution patterns were distinct between the two hydrological years of the study (Fig. 1a, Table S1). In the first year of the study (2004), a wet autumn and winter was followed by a dry spring (51 % of the long-term average precipitation). In contrast, in the second year (2005), a typically wet autumn preceded an extremely dry winter (only 27 % of the long-term average precipitation) and a moderately dry spring (79 % of the long-term average precipitation). Total precipitation was 607 mm in 2004 and 410 mm in 2005. Indeed, 2005 was the driest year in the last 140 years in the

southwestern Mediterranean (García-Herrera et al., 2007; Caldeira et al., 2015). For this reason, the studied years will be named henceforth as mild and dry years, for 2004 and 2005, respectively. These two years were also contrasting regarding temperature (Fig. 1b). The mild year was warmer than the dry year (mean annual temperature was 16.3 and 15.3 °C, respectively; Table S1), mostly due to the occurrence of an unusually cold winter during the dry year. Compared to the long-term average, both years presented cooler autumns and winters and warmer springs and summers. From March to May, mean

air temperatures were higher in the dry year, as compared to the mild year, inverting in June. The degrees-day sum (DDS) until budburst were not significantly different among years (p>0.05, 461 ± 13 and 431 ± 19 °C for the mild and dry year, respectively).





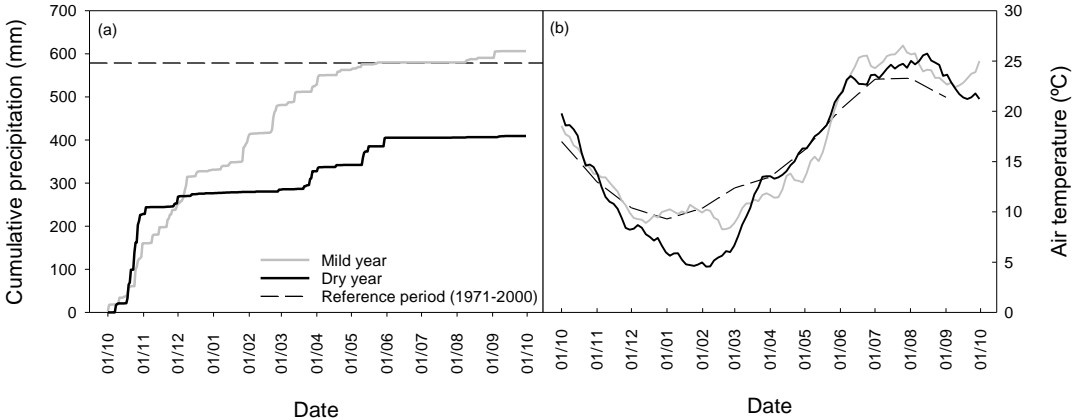

**Figure 1. (a)** Cumulative precipitation (mm) and **(b)** smoothed (7 days running average) daily mean temperature (ºC) for the study period. The long-term precipitation and monthly mean temperatures are included as a reference.

## 3.2 Phenological responses to drought

### 3.2.1 Phenophases timing and duration

The onset of the bud development and budburst were significantly delayed in the dry year (p<0.001) in comparison to the mild year, without a significant reduction of the bud development duration (p>0.05, $18 \pm 2$ and $16 \pm 2$ days for mild and dry years, respectively, Fig. 2). Shoot elongation, consequently, started later in the dry year (p<0.001) but ended earlier than in the mild year (p<0.001), leading to an important shortening of the shoot elongation period in ca. of 42 days (p<0.001, $66 \pm 2$ and $24 \pm 2$ days for mild and dry years, respectively; Fig. 2).

Significant differences were also observed in the timing and duration of trunk growth between years. Trunk growth started slightly later (p>0.05) and ended significantly earlier (p<0.001) in the dry year than in the mild year, thus showing a significantly shorter trunk growing period (p<0.001, $88 \pm 3$ and $57 \pm 2$ days for mild and dry years, respectively; Fig. 2). As a consequence, the spring growing season length (p<0.001, -48 %) was dramatically shortened. It is noteworthy that all trees ceased trunk growth within the same week in both years. The bulk of leaf senescence period was significantly delayed (p=0.031) and was longer in the dry year (p<0.05, $53.0 \pm 1.3$ versus $78.0 \pm 0.0$ days in the mild and dry years, respectively, corresponding to $50.9 \pm 2.3$ % and $76.1 \pm 2.6$ % of the total leaf senescence; Fig. 2 and Fig. 3).



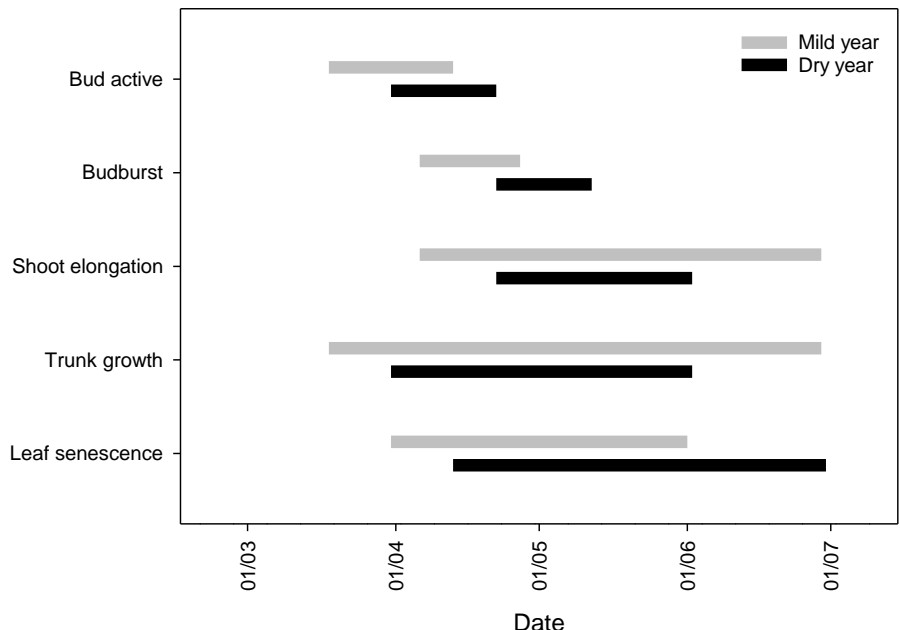

**Figure 2.** Timing of the onset and cessation of bud development (bud active), budburst, shoot elongation, trunk growth and leaf senescence in the two contrasting years. The bars represent the duration of each phenophases since the first until the last observation, in spring (n=15).

Along with the delay in the studied phenophases of the cork oak trees in the dry year, changes in their dynamics were also observed (Fig. 3). Budburst, although with a significant starting delay in the dry year, had similar development dynamics in both years (Fig. 3a). Shoot elongation, which started after budburst (Fig. 3b), had a faster development although in a shorter period of time in the dry year, with a fast buildup of new leaves (Fig. 3c). Spring trunk growth (BAI) exhibited a similar pattern in both years, although reaching significantly lower growth in the dry year ($p < 0.001$; Fig. 3d and Fig. 4c). Leaf senescence

was the most decoupled phenophase between years, with a later start in the dry year, but then progressing almost in parallel with the mild year (Fig. 3e). In the mild year, the bulk of leaf senescence finished before shoot elongation and trunk growth cessation. In contrast, in the dry year, the bulk senescence period was extended beyond the growing season (Fig. 2 and 3). In the dry year, the shoot elongation rate was significantly higher than in the mild year ($p < 0.001$, $0.39 \pm 0.07$ and $0.85 \pm 0.16$ cm day$^{-1}$ for mild and dry years, respectively). This latter phenophase occurred extremely concentrated in time. In turn, no changes

were observed in the rate of BAI ($p > 0.05$, $0.049 \pm 0.005$ and $0.050 \pm 0.006$ % day$^{-1}$ for mild and dry years, respectively) nor in leaf senescence rate ($p > 0.05$, $1.62 \pm 0.16$ and $1.69 \pm 0.12$ g m$^{-2}$ day$^{-1}$ for mild and dry years, respectively).





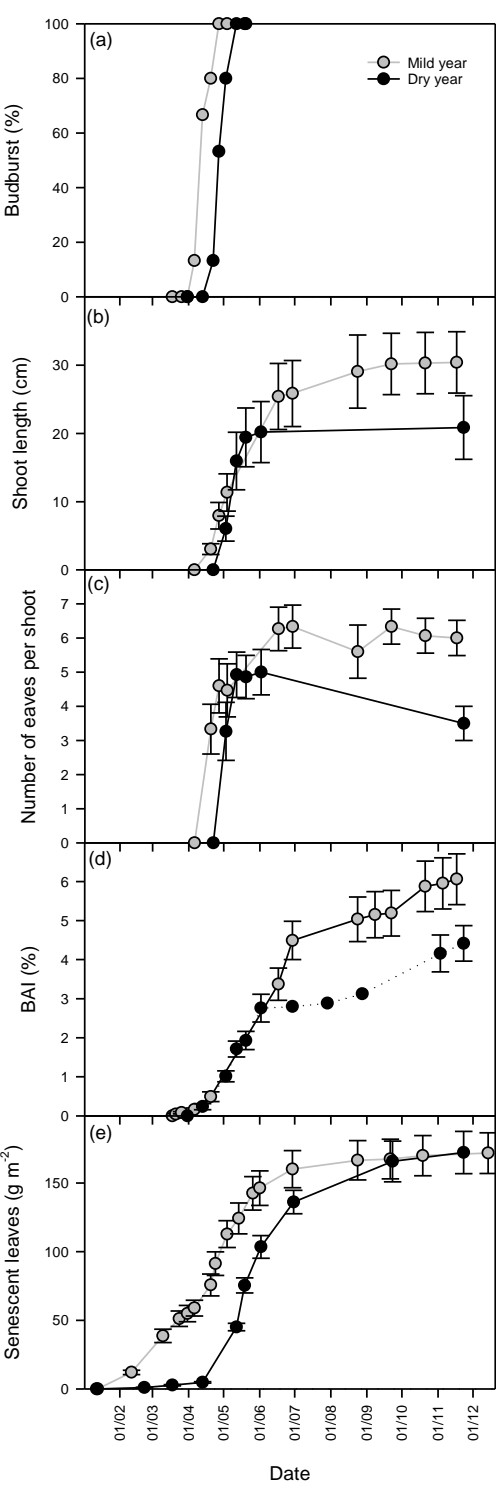





**Figure 3.** Growth dynamics in the mild and dry year, from January to December**. (a)** Budburst (%). **(b)** Shoot elongation (cm). **(c)** Number of leaves per shoot. **(d)** Basal area increment (BAI, cm$^2$). **(e)** Senescent leaves (g m$^{-2}$). Values are mean ± SEM (n = 15).

**3.2.2 Tree growth**

Overall tree growth was extremely hindered by drought (Fig. 3 and Fig. 4). Shoot elongation was significantly reduced in the dry year with new shoots being shorter (-22 %, p<0.05; Fig. 4a) and producing proportionally less leaves (-21 %, p<0.05; Fig. 4b) which was also corroborated by the absence of a significant difference in the internode length between years (p>0.05; Fig. S3). BAI was also significantly lower in the dry year, showing the highest reduction among the evaluated growth variables (-36 %, p<0.01; Fig. 4c). Leaf senescence was higher in the dry year (p<0.001; Fig. 4d), yet LAI was not affected (p>0.05, Fig. 4e). SLA was strongly reduced in the dry year (-23 %, p<0.0001; Fig. 4f).



**Figure 4.** Growth variables evaluated at the end of the growing season, in the mild and dry year. **(a)** Shoot length (cm). **(b)** Number of leaves per shoot. **(c)** Basal area increment (BAI, %) expressed as percentage of basal area at the beginning of each year. **(d)** Senescent leaves (g m$^{-2}$). **(e)** Leaf area index (LAI, leaves m$^2$ per ground m$^2$). **(f)** Specific leaf area (SLA, g cm$^{-2}$). Values are mean ± SEM (n = 15). Significant differences between years are indicated when * p<0.05, **p < 0.01, ***p < 0.001.



Shoot elongation period was highly and positively correlated with trunk growth period (r = 0.83, p<0.0001). Significant correlations were also found between BAI and shoot length, either when BAI was considered as a percentage of initial basal area (r = 0.43, p<0.05) or as absolute growth, in cm², (r = 0.57, p<0.001). Trunk growth duration was positively correlated

with spring BAI (%, r = 57, p<0.01). Shoot length was negatively correlated with SLA (r = 0.48, p<0.01).

### 3.3 Physiological status

Drought significantly decreased leaf water status (p<0.0001; Fig. 5a). Predawn leaf water potential ($\Psi_{pd}$) over the growing season was lower in the dry year than in the mild year, starting to decrease occurring earlier. Then, $\Psi_{pd}$ decreased in parallel,

as indicated by the absence of significant differences between regression slopes (p>0.05). Leaf carbon assimilation was also affected by drought (p<0.01; Fig. 5b). Maximal photosynthetic rate ($A_{max}$) was higher at the beginning of the growing season in both years, decreasing similarly as drought stress progressed (similar regression slope, p>0.05). Growth cessation, as indicated by the red asterisks in Fig. 5a, occurred at lower $\Psi_{pd}$ in the dry year than in the mild year. Indeed interpolated $\Psi_{pd}$ values were -0.84 and -1.33 MPa, for mild and dry year, respectively. Both values diverge in some degree from the previously

proposed threshold (-1.0 MPa) for growth cessation in cork oak (Pereira et al., 1987). Interestingly, the interpolated $A_{max}$ at growth cessation was similar in the studied years, as indicated by the red asteriscs in Fig. 5b.

Basal area increment was weakly but positively related with leaf carbon assimilation measured at the onset of the growing period (BAI (%) = -0.777 + 0.357 March $A_{max}$, r² = 0.18, p<0.05).



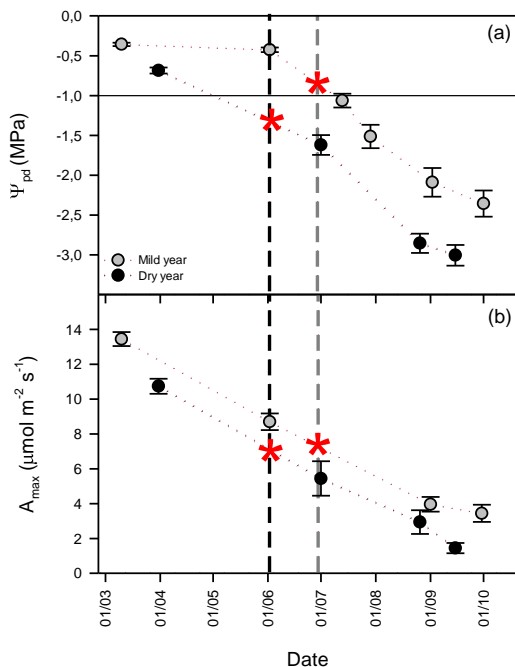

**Figure 5.** Tree **p**hysiological status over the growing season in the mild and in the dry year. **(a)** Predawn leaf water potential ($\Psi_{pd}$, MPa). **(b)** Maximal photosynthetic rate ($A_{max}$, µmol m$^{-2}$ s$^{-1}$). Vertical dashed lines represent the mean day of growth cessation for mild (grey) and dry (black) years, red asterics are indicative of the interpolated $\Psi_{pd}$ and $A_{max}$ at growth cessation. The horizontal black line is a reference for the reported threshold of $\Psi_{pd}$ (-1.0 MPa) for growth cessation in cork oak (Pereira et al., 1987). Values are mean ± SEM (n = 15).

### 3.4 Nitrogen resorption efficiency

Nitrogen concentration in green leaves was significantly different among years (p<0.01; Fig. 6a), mostly due to a significant reduction of [N$_{gr}$] at the onset of bulk leaf senescence period in the dry year. Mean [N$_{gr}$] was 16.1 ± 0.3 and 15.0±0.2 mg g$^{-1}$ in mild and dry year, respectively. The [N$_{se}$] decreased significantly in the dry year (p<0.01; Fig. 6b) and notably over the senescence period (p<0.001, see also Fig. S4 in which is noticeable the onset of leaf senescence with a steep decrease of [N$_{se}$] over time). The observed changes in [N$_{se}$] led to a significant increase of NRE, from the onset to the end of leaf senescence period (p<0.001) and, more importantly, in response to drought (+22 %, p<0.01; Fig. 6c).

NRE, considered globally, was only highly explained by [N$_{se}$] (NRE = 97.19 - 6.13 [Nse], r$^2$ = 0.97, p<0.0001; Fig. S5b). All observations at the end of senescence of the dry year dropped to values around 5 mg g$^{-1}$ (5.27 ± 0.12 mg g$^{-1}$) which might suggest a possible threshold for nutrient resorption proficiency or potential resorption (Killingbeck, 1996).





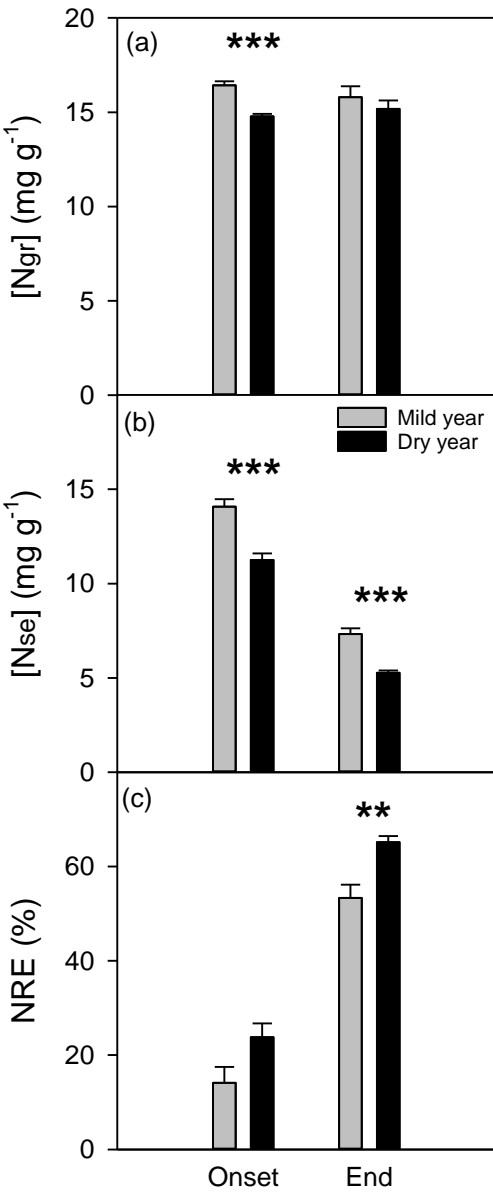

**Figure 6.** Nitrogen dynamics evaluated at the onset and end of bulk leaf senescence period in the mild and dry year. **(a)** Nitrogen concentration in green leaves ([Ngr], mg g$^{-1}$). **(b)** Nitrogen concentration in senescent leaves ([Nse], mg g$^{-1}$). **(c)** Nitrogen resorption efficiency (NRE, %). Values are mean ± SEM (n = 6). Significant differences between years area indicated when * $p < 0.05$, ** $p < 0.01$, *** $p < 0.001$.



## 4 Discussion

### 4.1 Drivers of spring growing season onset, cessation and length

The timing of the budburst of cork oak trees (Fig. 2) was determined by the accumulated temperature above a certain threshold (*i. e.* DDS) as previously shown in other studies (e.g. Pinto et al., 2011; Sampaio et al., 2016). The DDS until budburst was

not significantly different among years, although the timing of budburst was delayed in the dry year. This delay was probably due to the colder temperatures observed during the winter of the dry year, which just allowed the trees to reach the appropriate thermal time after the increase in temperature in the early spring. The importance of temperature on budburst, either cumulative (DDS) and/or early spring temperature, has been highlighted in several studies (e.g. Menzel et al., 2006; Richardson et al., 2006; Vitasse et al., 2009; García-Mozo et al., 2010; Körner and Basler, 2010; Pinto et al., 2011). Pinto et al. (2011) reported

that the only variable related to budburst date of adult cork oaks was air temperature, in particular average daily temperature 1.5 months prior to budburst. In a study across Europe, the timing of budburst was found to be highly correlated with the temperature of the preceding month (Menzel et al., 2006) and in a hardwood forest in northern USA, DDS accounted for more than 90% of the variation in spring canopy development (Richardson et al., 2006). In a cork oak provenances study (Sampaio et al., 2016), apart from a strong effect of the provenance origin on the timing of budburst, DDS was the strongest

environmental determinant of the timing of budburst between different years. Besides, the similar budburst dates found in our study and for the cork oak Portuguese provenances in the Sampaio et al. (2016) study, for the same mild (DOY 107 and 105, respectively) and dry years (DOY 120 and 113, respectively), was also noticeable.

The impact of concurrent drought on tree phenology is scarcely mentioned in the literature and it seems to indicate only a minor and not significant effect of drought on budburst timing of Mediterranean trees (Ogaya and Penuelas, 2004; García-

Mozo et al., 2010). Our results are in agreement with these studies, showing the major effect of temperature on the onset of phenological development, as well as the importance of spring temperatures to fulfill the temperature requirements for budburst (Fig. S1). Trunk growth started concomitantly with bud activity (Fig. 2 and Fig. 3), when temperatures increased, as was already shown for cork oak trees (Oliveira et al., 1994), and for other Mediterranean trees (Lempereur et al., 2015).

Cumulative precipitation was significantly lower in the dry year (Fig. 1a), actually the driest year in the last 140 years in the

southwestern Mediterranean (García-Herrera et al., 2007; Caldeira et al., 2015). Water availability strongly constrained the duration of shoot elongation and trunk growth (Fig. 2), thus reducing the length of the growing season. In agreement with our results, a shoot growing period of 10 weeks was reported, in an average precipitation year, for juvenile cork oak trees (Pereira et al., 1987). The growing season length in the dry year, as compared to the mild year, was more determined by an earlier cessation of growth (-30 days) than by a delayed budburst (+ 15 days). Shoot elongation and trunk growth ceased earlier in the

dry year and, interestingly, in a very synchronously way between trees in both years, as also reported by Oliveira et al. (1994). This observed synchrony suggests a trigger for growth cessation. A threshold of -1.0 MPa of leaf water potential, which was considered to be the onset of severe plant water stress, has been suggested to trigger growth cessation in cork oak (Pereira et



al., 1987). A threshold of -1.1 MPa was recently suggested for growth cessation of *Q. ilex* (Lempereur et al., 2015). In our study, growth ceased at interpolated $\Psi_{pd}$ values of -0.84 and -1.33 MPa, for mild and dry year, respectively, which suggest some acclimation to the precipitation conditions of the year. As $\Psi_{pd}$ reflects soil water availability for trees, it integrates all the water received by the system and appears to be a suitable trigger of growth cessation in water-limited ecosystems, as already

suggested by Pinto et al. (2011). In our study, growth ceased at significantly lower $\Psi_{pd}$ in the dry year as compared the mild year (Fig 5a), but at a similar carbon assimilation rate (Fig. 5b). This results call attention to a possible carbon limitation for growth, which occurred earlier in the dry year yet at higher $\Psi_{pd}$ in the mild year.

Climate has been shown to account for more than 80% in the variability of budburst day and growing season length in Mediterranean plants, mostly due to temperature, while precipitation only accounted for less than 10% in that variability

(Gordo and Sanz, 2010). However, in our study, water availability played a decisive role on the growing season length. This is most probably explained due to the cork oak phenology as leaf renewal (budburst and leaf senescence) occurs in a short period during spring/early summer, contrary to the more common pattern of spring budburst to autumn leaf senescence (Gordo and Sanz, 2010). Late spring/early summer corresponds to the onset of the drought season that has recently been anticipated and aggravated by the ongoing climatic changes, critically affecting the functioning of trees.

The leaf senescence period was reported to be smaller than shoot growth period (Pereira et al., 1987), as was observed in the mild year (Fig. 2), besides some occurrence registered before the bulk period (Fig. 3), probably due to strong rain events. In the dry year, the bulk of leaf senescence was the longer phenophase (Fig. 2), ceasing later than shoot or trunk growth. Losing more leaves for a longer period may be a response to drought, as by shedding older leaves, trees prevent water loss (Pereira et al., 2009) and/or as a nitrogen conservation strategy. The lower concentration of nitrogen in senescent leaves observed, at the

end of bulk period in the dry year, suggest that trees were able to recycle N likely by shedding leaves for a longer period. By increasing NRE, hence increasing N concentration in current year leaves, trees may have mitigated the negative effects of drought on N acquisition (Fig. S4).

### 4.2 Impact of drought on spring tree growth

Mediterranean evergreen trees take advantage of every favorable environmental conditions for carbon assimilation and growth (Larcher, 2000). Our data clearly show this, as trees had higher carbon assimilation rates (Fig. 5) while water was still not limiting, as indicated by predawn leaf water potential. Drought, by first inhibiting cell growth, affects primarily growth (Taiz and Zeiger, 1998). Stomatal closure occurs next, limiting carbon assimilation while preventing water loss (Chaves, 1991; Chaves et al., 2016). A strong stomatal control is an important feature of cork oak to withstand long periods of water shortage

and is closely linked to carbon assimilation (Otieno et al., 2007; Grant et al., 2010; Vaz et al., 2010; Pinto et al., 2012; Besson et al., 2014), corresponding to a drought avoidance, or isohydric, behavior. Despite the significant relationship found between



$A_{max}$ at the onset of growth and spring BAI, which reflects the importance of short-term weather conditions for carbon assimilation and growth, $A_{max}$ only explained 18% of the variability observed in BAI. The weakness of the relationship reflects the allocation of carbon not only for trunk growth but also to roots and canopy. In fact, a deep rooting system is of major importance for cork oak during summer and constitutes a considerable sink of carbon reserves (Mendes et al., 2016).

Growth was intrinsically linked to phenology dynamics, in which the duration of the phenophases exerted a major control (Fig. 3). This was particularly true regarding trunk growth, given the same growth rate over a shorter period in the dry year (Fig. 3d). In a drought manipulation study with *Q. ilex*, growth duration was the best predictor of BAI (Lempereur et al., 2015). In our study, spring BAI (%), besides being also positively correlated with trunk growth duration, was reduced roughly in the same proportion than precipitation reduction, when comparing the dry year with the mild year (-36% and -30 %, for BAI and

precipitation, respectively). In turn, trunk growth duration and shoot elongation duration were positively correlated, as well as BAI with shoot length. Nevertheless, shoot elongation was less inhibited by drought (-22 %) than BAI. The shoot length in our study, despite the severe drought observed, was considerably higher than what was reported in another study with juvenile cork oak trees (Oliveira et al., 1994), in which shoot length ranged between 1.8 and 6.2 cm yr$^{-1}$. This has probably to do with the fact that the sampled branches in that study were not clearly defined as apical, like we measured, thus showing lower

growth. The lower reduction in shoot elongation, as compared to trunk growth, combined with an absence of significant changes in LAI suggests that cork oak privileges leaf carbon assimilation, by maintaining the leaf area, as suggested elsewhere (Costa-e-Silva et al., 2015).

Although reported LAI or litterfall responses to drought are conflicting (Reichstein et al., 2002; Ogaya and Penuelas, 2004; Limousin et al., 2009; Martin-StPaul et al., 2013; Costa-e-Silva et al., 2015), we would expect a reduction in LAI as a

consequence of lower shoot elongation and lower number of leaves, without a corresponding decrease in total litterfall. Despite the higher leaf senescence observed during the bulk period, in the dry year, total litterfall (from January to September) was similar among years (Fig 3e). This could be explained by trees retaining a portion of leaves in the canopy longer than a year or simply because litterfall reflects the previous year growth that was not affected by drought. In a study realized in the Mediterranean region during the same years, also no changes were observed in *Q. ilex* litterfall (Limousin et al., 2009).

The observed reduction of SLA is a characteristic response of Mediterranean trees to drought (Chaves and Oliveira, 2004). By limiting the transpiring area, trees will be able to keep photosynthetic activity for a longer period, as suggested by Ramírez-Valiente et al. (2010), explaining the negative correlation between SLA assessed prior to the onset of drought and shoot length in cork oak, as we also observed. More schlerophyllous leaves were also associated with low soil nutrient availability (Salleo and Nardini, 2000), due to its higher ratio of carbon to nitrogen.

The short-term interactions between precipitation and temperature are complex (Gerst et al., 2017; Lempereur et al., 2017). The higher rate of shoot elongation observed in the dry year and a longer duration of shoot elongation in the mild year suggest,





respectively, a positive effect temperature early in the growing season but a negative or null effect during the elongation period in the mild year. A longer study period would be required to get reliable conclusions.

Cork oak trees "used the water sparingly" (Pereira et al., 2009) and the spring precipitation was of utmost importance for spring growth in the dry year (Fig. 1), which is in agreement with other studies (Besson et al., 2014; Costa et al., 2016; Oliveira et al., 2016). In contrast, for the mild year, the cumulated precipitation allowed for a longer growing period and consequently for a higher tree growth, irrespective of a warmer and drier spring.

## 4.3 Nitrogen contents and resorption efficiency

A decrease in N availability is an expected consequence of drought (Sardans et al., 2008), which might be aggravated by the forecasted changes in precipitation regimes (Rodríguez et al., 2019), leading to decreased [$N_{gr}$] (Fig. 6a) (Delarco et al., 1991). Nitrogen concentration in green leaves in the mild year was is same range than other studies (Delarco et al., 1991; Escudero et al., 1992; Oliveira et al., 1996; Kattge et al., 2011) but Andivia et al. (2009) reported a higher variation in [$N_{gr}$] over the growing season.

Nitrogen concentration in senescent leaves [$N_{se}$], known as resorption proficiency (Killingbeck, 1996) decreased steeply from the onset to the end of the bulk period of leaf senescence (Fig 6b, Fig. S4). Silla and Escudero (2003), studying the internal N cycling of *Q. ilex*, observed that after budburst, plants firstly relied on soil uptake and later on N resorption to supply growth requirements, which might explain our results. A more simple explanation lies on the fact that the first leaves shed by the tree were not senescent yet, thus N was incompletely resorbed from them.

More importantly, [$N_{se}$] decreased significantly in the dry year, reaching minimum values around 5 mg g$^{-1}$ (5.27 ± 0.12 mg g$^{-1}$). Minimum reported resorption proficiency for oaks was 4.5 mg g$^{-1}$ and resorption was considered highly proficient when [$N_{se}$] was lower than 7 mg g$^{-1}$ (Killingbeck, 1996). According to the exposed, the resorption observed in cork oak was highly proficient and the lowest values of [$N_{se}$] observed at the end of the senescence period, in the dry year, suggest the potential resorption for cork oak trees.

An increase in nitrogen resorption from senescent leaves is a recognized feature of Mediterranean plants (Andivia et al., 2009; Sardans and Peñuelas, 2013) and is a key feature of nutrient conservation strategies (Vergutz et al., 2012; Brant and Chen, 2015; Xu et al., 2017). Cork oak has been reported to resorb N more than the average in Mediterranean woody species (Delarco et al., 1991). The NRE (63 %) in the dry year was far higher than the 47 % or 56 % that were reported for evergreen trees (Aerts, 1996; Vergutz et al., 2012) or even for the 48 % or 52% for cork oak (Delarco et al., 1991; Escudero et al., 1992), the latter being similar to NRE in the mild year (54 %).

Cork oak increased significantly NRE (+22 %) with drought (Fig. 6c) and, by doing this, mitigated the effects of drought on N uptake from soils, as [$N_{gr}$] at the end of the growing period was similar among years (Fig. 6a). We found no general



correlation between [$N_{gr}$] and [$N_{se}$] nor between NRE and [$N_{gr}$] during leaf senescence. NRE was only significantly and negatively correlated with [$N_{se}$]. These findings underline the lack of nutritional control on NRE (Aerts and Chapin, 1999), as previously reported (Delarco et al., 1991; Aerts, 1996; Kazakou et al., 2007), despite recent studies have identified leaf nutrient status as the major control of NRE (Kobe et al., 2005; Vergutz et al., 2012; Zhao et al., 2017). Rather, the major control on

NRE was exerted by nutrient proficiency, as observed elsewhere (Kazakou et al., 2007). Although nutrient resorption can vary significantly from year to year (Killingbeck, 2004), has been considered a possible adaptive trait (Vergutz et al., 2012) . Our data confirm this last statement.

## 5 Conclusions

This study used field data to report changes in tree phenology in response to a naturally occurring extreme drought. As far as

we are aware, this study was the first to integrate the timing and duration of several phenophases with growth, while using green and leaf senescent to assess leaf nitrogen dynamics during the growing period. Our results confirmed temperature as a major driver of budburst, yet drought severely constrained tree growth mainly by strongly reducing the growing season length. Notably, the nitrogen resorption efficiency from senescent to green leaves increased as a response to drought. The different dynamics observed in shoot elongation, in contrast with the other phenophases, are indicative that cork oak privileges leaf area

development, while all other phases develop at the same rate. A higher nitrogen resorption efficiency in response to drought may clearly improve tree fitness in the short-term, but is likely to exert a negative feedback on the nitrogen cycle which might affect ecosystem functioning in the long-term under the forecasted increasing occurrence of droughts.

We are aware that the time frame of this study is narrow and longer studies are needed to confirm our results. We are also aware of the need, and our results support it, to understand the interactive effects of warming and heat waves with drought on

the phenology and growth of Mediterranean trees. Nevertheless, our study shows the dynamics of the different vegetative phenophases, which is not possible by ring-width analysis and contributes for a deeper understanding of the Mediterranean trees responses to drought. Finally, the sensitivity of phenology to climate change has implications for land management, including adaptation measures (Richardson et al., 2013; Han et al., 2018) to mitigate the deleterious effects of climate change. Our results corroborate the utility of using tree water balance as a management tool, as proposed by Cabon et al. (2018).

*Author contributions*. RLV performed fieldwork, data analysis, and wrote the manuscript. CKB planned the experiment and performed fieldwork. MCC assisted the interpretation of the data and contributed to manuscript writing. All authors critically discussed and reviewed the manuscript.

*Competing interests*. The authors declare that they have no conflict of interest.

*Acknowledgements:* This work is a product of a postdoctoral fellowship of Portuguese FCT to RLV (SFRH/BPD/86938/2012), and FCT project grant to CKB (PIEZAGRO, PTDC/AAG-REC/7046/ 2014). It was previously supported by EU project MIND-EVK2-CT-2002-00158 (Mediterranean terrestrial ecosystems and INcreasing Drought: vulnerability assessment). Centro de Estudos Florestais (CEF) and Instituto Dom Luiz are research units funded by FCT (UID/AGR/00239/2013 and UID/GEO/50019/2013, respectively). We are very grateful to Alastair Herd, André Pestana, Carla Nogueira, Elisabete Marianito, Elsa Breia, João Banza, and Pedro Almeida for their support for field
installation, site maintenance and/or field measurements.



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
