# Peer review of "Drought reduces tree growing season length but increases nitrogen resorption efficiency in a Mediterranean ecosystem"

_Biogeosciences, 2018_

## Referee Comment (RC1) · Anonymous Referee #1 · 1 Oct 2018

Dear editor The author Lobo-do Vale et al submitted manuscript- Drought reduces tree growing season length but increases nitrogen resorption efficiency on a Mediterranean ecosystem. This paper is interesting, well fall into the research scopes of Biogeosciences. But it is not prepared for this version, it need more effort to modify it for the whole paper. Now I will suggest major revision.

Dear author

Thanks for the author Lobo-do-Vale et al present the study entitled with Drought reduces tree growing season length but increases nitrogen resorption efficiency on a Mediterranean ecosystem. It is interesting and also crucial important to add the knowl-

edge on the influences by climate extremes (in particular for growing seasons) on Mediterranean ecosystems. The paper has shown a lot of information (phenophases such as bud development, budburst, shoot growth, and nitrogen resorption efficiency) based on the rich data collected in extreme drought years of 2005 and normal year of 2004. Personally, I quite like these topics and results, it well-fits for the research scopes of Biogeosciences. But there are a lot of descriptions and sentences are not clear for whole paper. Two main suggestions, one is to analyse the variables with the aridity index including temperature and precipitation together, such as SPEI or others. Second, the authors should take more time on the grammar, logic and structure for the whole paper.

Abstract Line 25 Please describe directly the sentence, such as the contrasting years of extreme drought in 2005 and moderate year in 2004. Line 15-18 Please short the sentence and represent your main purposes for this paper. Line 21 Is it -21% or 21%? Line 24 Is it +22% or 22% Which results are the crucial finding in your abstract, please highlight it.

Introduction Page 2 Line 6 Which ecosystem services, please add them. Line 7 Please add the details on the complex and highly variable Line 7-8 Are you sure all of the citations relate to this clarity. And the paper has stacked serval studies, please check the whole paper whether they are well-correlated to your sentence. Line14 Do you test the aridity including both temperature and precipitation, such as SPEI for the aridity site such as Mediterranean basin. Line 16-18 Although these studies are correlated with the clarity, need to reduce these citation number. Line 19-20, Which information is scarce?

Page 3 Line 15-27, Please add the aridity including temperature and precipitation. Page 7 Line 26 Is it $p<0.05$? please check them whole paper. Line 21-22 I do not find the two years are contrasting, but similar for the temperature, please recheck. Page 8 Line 8 and 13 Is it $p>0.05$? check for the whole paper Page 13 Do you have the graph or data to show the correlation? If not, please add it. Line 14-15 Is it necessary to

discuss here, maybe better put it in discussion section. Line 17 If you used the BAI instead of basal area increment, please continue to use it after the first definition. Line 18 Please also show the regression in the supporting figure.

Page 14 Line 8 Do you compare the values between years? Why use "among"? Line 14 What it means for "considered globally", and there is no regression in the graph. Line 15-16 Why discuss the result not in the discussion section? Page 16 Line 5 Why use "among"? Line 8-9 Are you sure the citations mentioned the same results with yours?

Page 19 Line 11 This sentence is not understandable, please restructure it

Please also note the supplement to this comment:
https://www.biogeosciences-discuss.net/bg-2018-393/bg-2018-393-RC1-supplement.pdf

---

## Referee Comment (RC2) · Anonymous Referee #2 · 22 Oct 2018

This study aimed at untangling the effects of drought on timing, duration and amount of growth, budburst, and nitrogen resorption efficiency. For this, the authors used 2 subsequent years, one of which was a mild drought, the other a strong drought. The authors found significant reductions in timing and duration of phenological parameters like budburst and growth, and found an increased nitrogen resorption in drought stressed trees, mitigating the negative effects of drought on N uptake from the soil.

The authors study some important aspects of effects of a changing climate: the phenology of trees. In the discussion they are however a bit chaotic, and strong explanations on what has been caused by drought and what could be caused by other factors are

partly there but not very elaborate.

I think I understand that only dry/ambient trees were used, and not the wet trees. This has to be more clearly explained though, even if referred to another paper, it would be nice to have a clear explanation in this paper on the exact study sample. Why were the wet trees not used? That could have been a nice interaction: drought years and trees growing with more or less water.

I advice to revise the discussion. It is a bit chaotically structured. Different measured variables come back every time with another focus, but then there is also overlap (trunk growth first occurs in terms of timing and duration (p16, line 25 a.f), then later it comes back again in growth rate but also again duration (p18 line 5 etc)).

The authors mention the short time frame of their study in the conclusion, but it would be nice to already discuss some of this in the discussion. Because, what could be the effect of time lags of drought on growth? Are there any legacy effect?

Some minor comments

Could you indicate significance in figure 2?

I feel that figure 4 is unnecessary next to figure 3. If significance would be indicated in figure 3, all those relations would already be shown there.

P14 line 16, move to discussion

Discussion P16 line 1-7 What was the DDS at budburst in this study? This does not have to be speculated on, the data is there to calculate, right?

P17 line 1-7 Is water the only factor that can cause cessation of growth? In the mild year, I think the cessation was not caused by a low water potential, but by other factors that determine the end of a growing season.

P20 line 9 In my view it has been speculated upon but not shown.

---

## Author Comment (AC1) · 28 Nov 2018

Responses to Reviewer 1

We thank the reviewer for the positive comments to the paper and suggestions that have improved the manuscript.

Dear author Thanks for the author Lobo-do-Vale et al present the study entitled with Drought reduces tree growing season length but increases nitrogen resorption efficiency on a Mediterranean ecosystem. It is interesting and also crucial important to add the knowledge on the influences by climate extremes (in particular for growing

seasons) on Mediterranean ecosystems. The paper has shown a lot of information (phenophases such as bud development, budburst, shoot growth, and nitrogen resorption efficiency) based on the rich data collected in extreme drought years of 2005 and normal year of 2004. Personally, I quite like these topics and results, it well-fits for the research scopes of Biogeosciences. But there are a lot of descriptions and sentences are not clear for whole paper. Two main suggestions: - one is to analyse the variables with the aridity index including temperature and precipitation together, such as SPEI or others

Answer: The SPEI index in our study can only be used to characterize the two climatologically contrasting years, because each year can only be associated to one unique SPEI value. However, for reference we have now included the SPEI computed on a 6-month period prior to growing season (March). SPEI was, respectively for 2004 and 2005, -0.4 (mild year) and -1.7 (dry year). We also added a reference on SPEI. Page 5 lines 3-6 and Page 7 lines 19-21.

- Second, the authors should take more time on the grammar, logic and structure for the whole paper.

Answer: We thank the reviewer. The paper was thoroughly checked and improved. A version with tracked changes is attached.

Abstract - Line 5: Please describe directly the sentence, such as the contrasting years of extreme drought in 2005 and moderate year in 2004.

Answer: We modified the sentence as: "In this study, we compared the spring phenology of cork oak trees (Quercus suber), an evergreen species, over two contrasting years, a mild year (2004) and dry year (2005), which was the most severe drought since records exist.". Page 1 lines 14-16

- Line 15-18: Please short the sentence and represent your main purposes for this paper. Answer: We modified the sentence as: "We evaluated the timing of occurrence,

BGD
duration, and intensity of bud development, budburst, shoot elongation, trunk growth, and leaf senescence (phenophases) and assessed the nitrogen resorption efficiency from senescent to green leaves." Page 1 lines 16-17

- Line 21: Is it -21% or 21%?

Answer: We modified the sentence as: "Shoot elongation was also reduced (-21 %)...". Page 1 line 20

- Line 24: Is it +22% or 22%

Answer: Our option to use +22% was to emphasize the increase in NRE. Page 1 line 23

- Which results are the crucial finding in your abstract, please highlight it.

Answer:. Abstract was improved to highlight our findings

Introduction Page 2 - Line 6: Which ecosystem services, please add them.

Answer: We corrected the sentence because the cycling of carbon, water and nutrients are also ecosystem services. The sentence was modified as follow: "Shifts in plant phenology have broad consequences for ecosystems, not only directly through changes in the cycling of carbon, water and nutrients, but also indirectly through feedbacks on the climate system (Diez et al., 2012; Richardson et al., 2013; Tang et al., 2016; Han et al., 2018)., which in turn may strongly influence ecosystem productivity (Tang et al., 2016)". Page 2 lines 4-7

- Line 7: Please add the details on the complex and highly variable

Answer: We corrected the sentence to be more specific: "However, the response of species phenology to climate change is complex and highly variable, depending on species, spatiotemporal scales (seasonal or annual), or main climate driver (precipitation, temperature or both) (Penuelas et al., 2004; Polgar and Primack, 2011; Diez et al., 2012; Tang et al., 2016; Gerst et al., 2017)". Page 2 lines 7-9
- Line 7-8: Are you sure all of the citations relate to this clarity. And the paper has stacked serval studies, please check the whole paper whether they are well-correlated to your sentence.

Answer: All the citations and references were confirmed. We modified the sentence to:"It is critical to have a better understanding of the patterns and climate drivers of the different phenophases in a broader range of tree species (Lempereur et al., 2017; Tang et al., 2016; Polgar and Primack, 2011; Richardson et al., 2013; Doblas-Miranda et al., 2015; )". Page 2 lines 10-11.

- Line14: Do you test the aridity including both temperature and precipitation, such as SPEI for the aridity site such as Mediterranean basin.

Answer: As was mentioned above, the SPEI index in our study can only be used to characterize the aridity of the two years, because each year can only be associated to one unique SPEI value. We now refer the two values of SPEI for both years. Page 7 lines 19-21

- Line 16-18: Although these studies are correlated with the clarity, need to reduce these citation number.

Answer: Although each citation was quite interesting, we reduced the citation number. Page 2 line 17

- Line 19-20: Which information is scarce?

Answer: Information of the impact of precipitation changes on tree phenology. We corrected the sentence as following: "Still, available information on the effects of precipitation on phenology is scarce". Page 2 line 19

Page 3 - Line 15-27: Please add the aridity including temperature and precipitation.

Answer: There is no information available, as far as we are aware, relating directly to the effects of aridity on NRE.

BGD
Page 7 - Line 26: Is it p<0.05? please check them whole paper.

Answer: The results indicated that the test result was not significant, thus p>0.05 is right. The budburst occurred at similar DDS in the two years, but later in the dry year because the winter was cooler. Now line 27

- Line 21-22: I do not find the two years are contrasting, but similar for the temperature, please recheck.

Answer: We deleted the words "These two years were also contrasting" because in fact only winter of the dry year was colder and the other seasons had similar temperature records.

Page 8 - Line 8 and 13: Is it p>0.05? check for the whole paper

Answer: The results indicated that the test result was not significant, thus p>0.05 is right. Bud development showed similar duration in both years as well as the onset of the trunk growth.

Page 13 - Do you have the graph or data to show the correlation? If not, please add it.

Answer: Figure S4 was added to supplemental material regarding the correlations mentioned in Page 13 lines 1-6.

- Line 14-15: Is it necessary to discuss here, maybe better put it in discussion section.

Answer: Those lines were eliminated, as the information was also provided in the discussion section, Page 17 lines 7-12.

- Line 17: If you used the BAI instead of basal area increment, please continue to use it after the first definition.

Answer: This has been done on Page 13 line 17.

- Line 18: Please also show the regression in the supporting figure.

Answer: This has been done on Page 13, line 17. The supporting figure was added to
supplemental material, Figure S5 and referred in text: "BAI was weakly but positively related with Amax at the onset of the growing period (BAI (%) = -0.777 + 0.357 March Amax, r2 = 0.18, p

Page 19 - Line 11: This sentence is not understandable, please restructure it

Answer: The sentence has been corrected as "(Delarco et al., 1991). [Ngr] in the mild year, was within the range of values observed in other studies (Delarco et al., 1991; Escudero et al., 1992; Oliveira et al., 1996; Kattge et al., 2011). Nonetheless, Andivia et al. (2009) reported a higher variation in [Ngr] over the growing season.", Page 19 lines 9-11

Please also note the supplement to this comment: https://www.biogeosciences-discuss.net/bg-2018-393/bg-2018-393-AC1supplement.zip

---

## Author Comment (AC2) · 28 Nov 2018

Response to reviewer 2

We thank the reviewer for the positive comments to the paper and suggestions that have improved the manuscript.

This study aimed at untangling the effects of drought on timing, duration and amount of growth, budburst, and nitrogen resorption efficiency. For this, the authors used 2 subsequent years, one of which was a mild drought, the other a strong drought. The authors found significant reductions in timing and duration of phenological parameters like budburst and growth, and found an increased nitrogen resorption in drought stressed trees, mitigating the negative effects of drought on N uptake from the soil. The authors study some important aspects of effects of a changing climate: the phenology of trees. - In the discussion they are however a bit chaotic, and strong explanations on what has been caused by drought and what could be caused by other factors are partly there but not very elaborate.

Answer: Discussion was re-structured and improved.

- I think I understand that only dry/ambient trees were used, and not the wet trees. This has to be more clearly explained though, even if referred to another paper, it would be nice to have a clear explanation in this paper on the exact study sample. Why were the wet trees not used? That could have been a nice interaction: drought years and trees growing with more or less water.

Answer: In the manuscript by Kurz-Besson et al. 2014, we showed a significant effect of the irrigation treatment on tree growth, while no significant effect of rainfall exclusion could be detected throughout the study period between 2003 and 2005, between trees from the DRY (rainfall excluded) and the AMBIENT (control) treatments. Therefore, we considered trees from the DRY and AMBIENT treatment as a single pool belonging to the same statistical population. Also the irrigation treatment only had a significant effect after a single pulse irrigation in late spring applied as a punctual experiment, that we did not consider representative of the rest of the irrigation performed in 2005. This is why we excluded trees from the WET treatment in our manuscript. We added an explanation to the manuscript Page 4 lines 15-21.

- I advice to revise the discussion. It is a bit chaotically structured. Different measured variables come back every time with another focus, but then there is also overlap (trunk growth first occurs in terms of timing and duration (p16, line 25 a.f), then later it comes back again in growth rate but also again duration (p18 line 5 etc).

Answer: Discussion was re-structured.

- The authors mention the short time frame of their study in the conclusion, but it would be nice to already discuss some of this in the discussion. Because, what could be the effect of time lags of drought on growth? Are there any legacy effect?

Answer: In fact, we can not exclude any legacy effect of previous droughts on tree phenology. However, because we observed a fully recovery on tree physiology shortly after the start autumn rains, as reported in Besson et al. 2014, we believe that the current hydrological year, or short-term environmental conditions, were the major drivers of phenology in the spring.

Some minor comments - Could you indicate significance in figure 2?

Answer: Because figure 2 shows the onset and cessation and, thus, duration of the phenophases, we believe that adding significances to the figure would make it difficult to interpret. Alternatively, we added a table (Table S2) to the supplemental material.

- I feel that figure 4 is unnecessary next to figure 3. If significance would be indicated in -figure 3, all those relations would already be shown there.

Answer: In Fig. 3 we want to show the dynamics of different phenophases over time. In Fig. 4 we want to emphasize the differences in spring growth. So, time scales considered in Fig. 3 and Fig. 4 are different. Pooling data in Fig. 4 allowed the detection of stronger significant differences leading to stronger conclusions for our manuscript.

- P14 line 16, move to discussion

Answer: In fact this information was already in the discussion section, so this sentence was deleted.

Discussion - P16 line 1-7 What was the DDS at budburst in this study? This does not have to be speculated on, the data is there to calculate, right?

Answer: On page 7 lines 27-28: "The degrees-day sum (DDS) until budburst were not significantly different between years ($p > 0.05$, $461 \pm 13$ and $431 \pm 19\ ^{\circ}$C for the mild

and dry year, respectively, Fig. S1b).

- P17 line 1-7 Is water the only factor that can cause cessation of growth? In the mild year, I think the cessation was not caused by a low water potential, but by other factors that determine the end of a growing season.

Answer: We can not exclude the possibility of an effect of VPD, that was higher in the mild year. Nevertheless, high VPD's were also registered in some periods during the growing season. We added some comments in the discussion, Page 17 line 13.

- P20 line 9 In my view it has been speculated upon but not shown. We have shown a clear effect of drought in shortening the spring growing season length, mostly by an early cessation of growth, with a correspondent decrease in growth variables, such as shoot elongation and trunk growth.

Please also note the supplement to this comment:
https://www.biogeosciences-discuss.net/bg-2018-393/bg-2018-393-AC2-supplement.zip

---

## Author Response (AR1)

**We thank the Editor and Reviewers for the positive and constructive comments on the manuscript that we have now included and addressed point-by-point below (responses in blue).**

**Responses to Editor**

Dear authors,

both reviewers agreed that the manuscript presents an interesting and large data set on the phenology of trees in response to natural droughts. While this is a timely topic, both reviewers also indicated that the manuscripts requires various clarifications, which you addressed in your responses. The reviewers also indicated that the discussion was not clearly structured. You have largely rewritten the discussion, and thus, the reviewers should again evaluate the manuscript.

In addition, I want to ask the authors to clarify how NIRS which has been used to determine N contents has been calibrated against measured N concentration (e.g. using a C/N analyzer).

*We added the NIR calibration details to section 2.5. Page 6, lines 19-26.*

Please increase the font size in the captions and legends of the Figures.

*Thank you for your suggestion. The font size in the captions and legends of the Figures was increased.*

**Responses to Reviewer 1**

Dear author

Thanks for the author Lobo-do-Vale et al present the study entitled with Drought reduces tree growing season length but increases nitrogen resorption efficiency on a Mediterranean ecosystem. It is interesting and also crucial important to add the knowledge on the influences by climate extremes (in particular for growing seasons) on Mediterranean ecosystems. The paper has shown a lot of information (phenophases such as bud development, budburst, shoot growth, and nitrogen resorption efficiency) based on the rich data collected in extreme drought years of 2005 and normal year of 2004. Personally, I quite like these topics and results, it well-fits for the research scopes of Biogeosciences.

*We thank the reviewer for the positive comments to the paper and suggestions that have improved the manuscript.*

But there are a lot of descriptions and sentences are not clear for whole paper. **Two main suggestions:**

- **one is to analyse the variables with the aridity index including temperature and precipitation together, such as SPEI or others**

*The SPEI index in our study can only be used to characterize the two climatologically contrasting years, because each year can only be associated to one unique SPEI value. However, for reference we have now included the SPEI computed on a 6-month period prior to growing season (March). SPEI was, respectively for 2004 and 2005, -0.4 (mild year) and -1.7 (dry year). We also added a reference on SPEI. Page 5 lines 3-6 and Page 7 lines 19-21.*

- **Second, the authors should take more time on the grammar, logic and structure for the whole paper.**

*We thank the reviewer. The paper was thoroughly checked and improved. A version with tracked changes is attached.*

**Abstract**

- Line 5: Please describe directly the sentence, such as the contrasting years of extreme drought in 2005 and moderate year in 2004.

*We modified the sentence as: "In this study, we compared the spring phenology of cork oak trees (Quercus suber), an evergreen species, over two contrasting years, a mild year (2004) and dry year (2005), which was the most severe drought since records exist.". Page 1 lines 14-16.*

- Line 15-18: Please short the sentence and represent your main purposes for this paper.

*We modified the sentence as: "We evaluated the timing of occurrence, duration, and intensity of bud development, budburst, shoot elongation, trunk growth, and leaf senescence (phenophases) and assessed the nitrogen resorption efficiency from senescent to green leaves.". Page 1 lines 16-17.*

- Line 21: Is it -21% or 21%?

*We modified the sentence as: "Shoot elongation was also reduced (-21 %)…". Page 1 line 20.*

- Line 24: Is it +22% or 22%

*Our option to use +22% was to emphasize the increase in NRE. Page 1 line 23.*

- Which results are the crucial finding in your abstract, please highlight it.

*Abstract was improved to highlight our findings.*

**Introduction**

- Line 6: Which ecosystem services, please add them.

*We corrected the sentence because the cycling of carbon, water and nutrients are also ecosystem services. The sentence was modified as follow: "Shifts in plant phenology have broad consequences for ecosystems, not only directly through changes in the cycling of carbon, water and nutrients, but also indirectly through feedbacks on the climate system (Diez et al., 2012; Richardson et al., 2013; Tang et al., 2016; Han et al., 2018)., which in turn may strongly influence ecosystem productivity (Tang et al., 2016)". Page 2 lines 4-7.*

- Line 7: Please add the details on the complex and highly variable

*We corrected the sentence to be more specific: "However, the response of species phenology to climate change is complex and highly variable, **depending on species, spatiotemporal scales (seasonal or annual), or main climate driver (precipitation, temperature or both**) (Penuelas et al., 2004; Polgar and Primack, 2011; Diez et al., 2012; Tang et al., 2016; Gerst et al., 2017)". Page 2 lines 7-8.*

- Line 7-8: Are you sure all of the citations relate to this clarity. And the paper has stacked serval studies, please check the whole paper whether they are well-correlated to your sentence.

*All the citations and references were confirmed. We modified the sentence to:"It is critical to have a better understanding of the patterns and climate drivers of the different phenophases in a broader range of tree species (Lempereur et al., 2017; Tang et al., 2016;  Richardson et al., 2013; )". Page 2 lines 9-10.*

- Line14: Do you test the aridity including both temperature and precipitation, such as SPEI for the aridity site such as Mediterranean basin.

*As was mentioned above, the SPEI index in our study can only be used to characterize the aridity of the two years, because each year can only be associated to one unique SPEI value. We now refer the values of SPEI for each year. Page 7 lines 19-21.*

- Line 16-18: Although these studies are correlated with the clarity, need to reduce these citation number.

*As suggested we reduced the number of cited references. Page 2 line 16.*

- Line 19-20: Which information is scarce?

*Information on the impact of precipitation changes on tree phenology. We corrected the sentence as following: "Still, available information on the effects of precipitation on phenology is scarce". Page 2 line 18.*

- Line 15-27: Please add the aridity including temperature and precipitation.

*There is no information available, as far as we are aware, relating directly to the effects of aridity on NRE.*

- Line 26: Is it p<0.05? please check them whole paper.

*The results indicated that the test result was not significant, thus p>0.05 is right. The budburst occurred at similar DDS in the two years, but later in the dry year because the winter was cooler. Page 7, line 28.*

- Line 21-22: I do not find the two years are contrasting, but similar for the temperature, please recheck.

*We deleted the words "These two years were also contrasting" because in fact only winter of the dry year was colder than the winter of the mild year. All the other seasons had similar temperature records. The sentence was corrected to "Regarding temperature (Fig. 1b), the mild year was slightly warmer than the dry year (mean annual temperature was 16.3 and 15.3 ºC, respectively; Table S1), mostly due to the occurrence of an unusually cold winter during the dry year.". Page 7, lines 23-25.*

- Line 8 and 13: Is it p>0.05? check for the whole paper

*As above, the results indicated that the test result was not significant, thus p>0.05 is right. Bud development showed similar duration in both years as well as the onset of the trunk growth. Page 8, lines 8 and 13.*

- Do you have the graph or data to show the correlation? If not, please add it.

*Figure S4 was added to supplemental material regarding the correlations mentioned in Page 13 lines 1-6.*

- Line 14-15: Is it necessary to discuss here, maybe better put it in discussion section.

*Those lines were eliminated, as the information was also provided in the discussion section, Page 18 lines 4-8.*

- Line 17: If you used the BAI instead of basal area increment, please continue to use it after the first definition.

*This has been done. Thank you. Page 14 lines 17-18 and though the whole paper.*

- Line 18: Please also show the regression in the supporting figure.

*This has been done. The supporting figure was added to supplemental material, Figure S5 and referred in text: "BAI was weakly but positively related with Amax at the onset of the growing period (BAI (%) = -0.777 + 0.357 March Amax, r2 = 0.18, p<0.05, Fig. S5)". Page 14 line 18*

- Line 8: Do you compare the values between years? Why use "among"?

*"Among" has been changed to "between" throughout the entire manuscript, e.g., Page 3 line 34, Page 7 line 28, Page 15 line 8, Page 18 line 3 and Page 19 line 24.*

- Line 14: What it means for "considered globally", and there is no regression in the graph.

*By "considered globally" we meant all data, or the overall correlation. The sentence was corrected to "Furthermore, NRE was closely explained by [Nse] (NRE = 97.19 - 6.13 [Nse], r2 = 0.97, p<0.0001; Fig. S5b)", Page 15 line 14. Also the regression was added to supplemental material, Figure S7b.*

- Line 15-16: Why discuss the result not in the discussion section?

*Those lines were eliminated, as the information was also provided in the discussion section, Page 20 lines 22-26.*

- Line 5: Why use "among"?
*As mentioned above, "among" has been changed to "between" throughout the entire manuscript, e.g., Page 3 line 34, Page 7 line 28, Page 15 line 8, Page 18 line 3 and Page 19 line 24.*

- Line 8-9: Are you sure the citations mentioned the same results with yours?
*We believe all the citations we provided are highly relevant in the field of phenology and well support our sentence, by showing an effect of temperature on leaf unfolding, budburst or the onset of vegetation greenness (for studies based on satellite data). We therefore kept the cited references. Page 17 lines 7-8.*

- Line 11: This sentence is not understandable, please restructure it

*The sentence has been corrected as "The variation of [Ngr] in the mild year, was within the same range of values than those observed in other studies (Delarco et al., 1991; Escudero et al., 1992; Oliveira et al., 1996; Kattge et al., 2011). Nonetheless, Andivia et al. (2009) reported a higher variation in [Ngr] over the growing season", Page 20, lines 14-16.*

**Responses to Reviewer 2**

This study aimed at untangling the effects of drought on timing, duration and amount of growth, budburst, and nitrogen resorption efficiency. For this, the authors used 2 subsequent years, one of which was a mild drought, the other a strong drought. The authors found significant reductions in timing and duration of phenological parameters like budburst and growth, and found an increased nitrogen resorption in drought stressed trees, mitigating the negative effects

of drought on N uptake from the soil. The authors study some important aspects of effects of a changing climate: the phenology of trees.

*We thank the reviewer for the positive comments to the paper and suggestions that have improved the manuscript.*

- **In the discussion they are however a bit chaotic, and strong explanations on what has been caused by drought and what could be caused by other factors are partly there but not very elaborate.**

*Discussion was re-structured and improved.*

- **I think I understand that only dry/ambient trees were used, and not the wet trees. This has to be more clearly explained though, even if referred to another paper, it would be nice to have a clear explanation in this paper on the exact study sample.** Why were the wet trees not used? That could have been a nice interaction: drought years and trees growing with more or less water.

*In the manuscript by Kurz-Besson et al. 2014, we showed a significant effect of the irrigation treatment on tree growth, while no significant effect of rainfall exclusion could be detected throughout the study period between 2003 and 2005, between trees from the DRY (rainfall excluded) and the AMBIENT (control) treatments. Therefore, we considered trees from the DRY and AMBIENT treatment as a single pool belonging to the same statistical population. Also the irrigation treatment only had a significant effect after a single pulse irrigation in late spring applied as a punctual experiment, that we did not consider representative of the rest of the irrigation performed in 2005. This is why we excluded trees from the WET treatment in our manuscript. We added an explanation to the manuscript Page 4 lines 13-19.*

- I advice to revise the discussion. It is a bit chaotically structured. Different measured variables come back every time with another focus, but then there is also overlap (trunk growth first occurs in terms of timing and duration (p16, line 25 a.f), then later it comes back again in growth rate but also again duration (p18 line 5 etc).

*Discussion was re-structured.*

- The authors mention the short time frame of their study in the conclusion, but it would be nice to already discuss some of this in the discussion. Because, what could be the effect of time lags of drought on growth? Are there any legacy effect?

*In fact, we cannot exclude any legacy effect of previous droughts on tree phenology. However, because we observed a fully recovery on tree physiology shortly after the start autumn rains, as reported in Besson et al. 2014, we believe that the current hydrological year, or short-term environmental conditions, were the major drivers of phenology in the spring.*

**Some minor comments**

- Could you indicate significance in figure 2?

*Because Fig. 2 shows the onset and cessation and, thus, duration of the phenophases, we believe that adding significances to the figure would make it difficult to interpret. Alternatively, we added a table (Table S2) to the supplemental material.*

- I feel that figure 4 is unnecessary next to figure 3. If significance would be indicated in -figure 3, all those relations would already be shown there.

*In Fig. 3 we want to show the dynamics of different phenophases over time. In Fig. 4 we want to emphasize the differences in spring growth. So, time scales considered in Fig. 3 and Fig. 4 are different. Pooling data in Fig. 4 allowed the detection of stronger significant differences leading to stronger conclusions for our manuscript.*

- P14 line 16, move to discussion

*In fact this information was already in the discussion section, so this sentence was deleted.*

Discussion

- P16 line 1-7 What was the DDS at budburst in this study? This does not have to be speculated on, the data is there to calculate, right?

*On page 7 lines 28-29: "The degrees-day sum (DDS) until budburst were not significantly different between years ($p > 0.05$, $461 \pm 13$ and $431 \pm 19$ ºC for the mild and dry year, respectively, Fig. S1b).*

- P17 line 1-7 Is water the only factor that can cause cessation of growth? In the mild year, I think the cessation was not caused by a low water potential, but by other factors that determine the end of a growing season.

*We cannot exclude the possibility of an effect of VPD, that was higher in the mild year. Nevertheless, high VPD's were also registered in some periods during the growing season (Fig. S8). We added some comments to the discussion, Page 18, line 14-18.*

- P20 line 9 In my view it has been speculated upon but not shown.

*We have shown a clear effect of drought in shortening the spring growing season length, mostly by an early cessation of growth, with a correspondent decrease in growth variables, such as shoot elongation and trunk growth.*

[revised manuscript text omitted]